# Endometrial PTEN Deficiency Leads to SMAD2/3 Nuclear Translocation

**DOI:** 10.3390/cancers13194990

**Published:** 2021-10-05

**Authors:** Núria Eritja, Raúl Navaridas, Anna Ruiz-Mitjana, Maria Vidal-Sabanés, Joaquim Egea, Mario Encinas, Xavier Matias-Guiu, Xavier Dolcet

**Affiliations:** 1Oncologic Pathology Group, Departament de Ciències Mèdiques Bàsiques, Institut de Recerca Biomèdica de Lleida, IRBLleida, Universitat de Lleida, Centro de Investigación Biomédica en Red Cáncer CIBERONC, 25198 Lleida, Spain; neritja@irblleida.cat (N.E.); raul.navaridas@udl.cat (R.N.); Anna.ruizmitjana@udl.cat (A.R.-M.); maria.vidalsabanes@udl.cat (M.V.-S.); fjmatiasguiu.lleida.ics@gencat.cat (X.M.-G.); 2Molecular Developmental Neurobiology Group, Departament de Ciències Mèdiques Bàsiques, Institut de Recerca Biomèdica de Lleida, IRBLleida, Universitat de Lleida, 25198 Lleida, Spain; joaquim.egea@udl.cat; 3Developmental and Oncogenic Signalling Group, Departament de Medicina Experimental, Institut de Recerca Biomèdica de Lleida, IRBLleida, Universitat de Lleida, 25198 Lleida, Spain; mario.encinas@udl.cat; 4Department of Pathology, Hospital Universitari de Bellvitge, 08908 Barcelona, Spain

**Keywords:** PTEN, TGF-β, SMAD2/3, endometrial cancer

## Abstract

**Simple Summary:**

PTEN is a protein highly altered in endometrial cancer. PTEN mutation or deficiency leads to the activation of other downstream proteins that are important to the development of cancers. In this study, we have identified the SMAD2/3 proteins as targets of PTEN deficiency. We have found that loss of PTEN in endometrial cells leads to SMAD2/3 activation. To investigate the role of SMAD2/3 activation downstream of PTEN deficiency, we have used endometrial cells lacking both PTEN and SMAD2/3 proteins. These cells display even more tumorigenic potential than cells lacking only PTEN. These results suggest that SMAD2/3 acts as an obstacle for cancer development triggered by PTEN loss.

**Abstract:**

TGF-β has a dichotomous function, acting as tumor suppressor in premalignant cells but as a tumor promoter for cancerous cells. These contradictory functions of TGF-β are caused by different cellular contexts, including both intracellular and environmental determinants. The TGF-β/SMAD and the PI3K/PTEN/AKT signal transduction pathways have an important role in the regulation of epithelial cell homeostasis and perturbations in either of these two pathways’ contributions to endometrial carcinogenesis. We have previously demonstrated that both PTEN and SMAD2/3 display tumor-suppressive functions in the endometrium, and genetic ablation of either gene results in sustained activation of PI3K/AKT signaling that suppresses TGF-β-induced apoptosis and enhances cell proliferation of mouse endometrial cells. However, the molecular and cellular effects of PTEN deficiency on TGF-β/SMAD2/3 signaling remain controversial. Here, using an in vitro and in vivo model of endometrial carcinogenesis, we have demonstrated that loss of PTEN leads to a constitutive SMAD2/3 nuclear translocation. To ascertain the function of nuclear SMAD2/3 downstream of PTEN deficiency, we analyzed the effects of double deletion PTEN and SMAD2/3 in mouse endometrial organoids. Double PTEN/SMAD2/3 ablation results in a further increase of cell proliferation and enlarged endometrial organoids compared to those harboring single PTEN, suggesting that nuclear translocation of SMAD2/3 constrains tumorigenesis induced by PTEN deficiency.

## 1. Introduction

TGF-β is a multimodal factor that participates in many biological and physiological processes. The variability of TGF-β functions is attributable to differences in cellular type and context [1]. TGF-β signaling pathways are triggered by its interaction to the TGF-β type II receptor (TGFβRII) that, in turn, interacts with the TGF-β type I receptor (TGFβRI or ALK5). TβRII phosphorylates TGFβRI and activates downstream effectors that transduce TGF-β signaling. The canonical TβRs signaling is conducted by the SMAD transcription factor family [2,3,4]. Engagement of TβR leads to the phosphorylation of the receptor-associated SMADs (R-SMADs), SMAD2 and SMAD3. Once phosphorylated SMAD2 and/or SMAD3 interact with the common SMAD (Co-SMAD) SMAD4, assembling dimers or trimers translocate to the nucleus. In the nucleus, SMAD4-R-SMAD bind other transcription factors that act as co-activators or co-repressors of transcription. A third group of SMADs are the inhibitory SMADs (I-SMADs) that compete with R-SMADs for receptor binding and by targeting activated receptor complex to proteasome degradation [5]. In addition to canonical SMAD signaling, TGF-β triggers other signaling pathways frequently referred as “non-SMAD” branch of TGF-β signaling [6,7]. These non-canonical TGF-β pathways include Rho-like GTPase signaling pathway, MAP kinase pathway and the Phosphatidylinositol-3 kinase/AKT (PI3K/AKT) signaling pathway.

In cancer development and progression, TGF-β has a dichotomous function, being a suppressor for premalignant or normal cells but a tumor promoter for transformed cells [8,9,10]. As a tumor suppressor, TGF-β elicits cell cycle inhibition and apoptosis, and loss of those responses are critical for cancer progression [9,11]. However, the mechanisms by which TGF-β switches its functions are not fully ascertained. An increasing amount of evidence demonstrates that tumor-suppressive signaling induced by TGF-β is impaired by oncogenic mutations, leading to survival and proliferation of initiated cells. Among such perturbations, those that activate the PI3K/AKT signaling pathway antagonize the cytostatic or pro-apoptotic effects of TGF-β [12].

The PI3K/AKT pathway regulates cell survival and proliferation and is frequently dysregulated in human cancers. PTEN (phosphatase and tensin homolog deleted on chromosome 10) is a phosphatase that opposes PI3K activity by dephosphorylating phosphatidylinositol-3,4,5-trisphosphate (PIP3) to phosphatidylinositol-4,5-trisphosphate (PIP2) [13]. Loss of PTEN activity is a frequent alteration in cancer, with special high incidence in endometrial cancer [14,15,16]. Alterations of PTEN increase the amount of PIP3 in the membrane, resulting in the activation of 3-phosphoinositide-dependent kinase (PDK) and AKT, which in turn stimulates cell proliferation and survival. The importance of PTEN deficiency in endometrial tumorigenesis has been evidenced by different knock-out mouse models, in which genetic deletion of PTEN results in the development of endometrial carcinogenesis [17,18,19].

The TGF-β/SMAD signaling pathway has an important role in the uterine function and physiology of the female uterine tract [20]. Genetically modified mouse models have uncovered the importance of TGF-β as a tumor suppressor in the female reproductive tract. Conditional TβRI knock-out in the female reproductive system shows profound defects in myometrium structure and function [21], and ablation of TβRI in the uterus leads to increased endometrial cell proliferation resulting in the development of endometrial hyperplasia and the development of endometrial cancers [22]. Moreover, uterine conditional deletion of TβRI [23], conditional double deletion of SMAD2 and SMAD3 [24] or conditional deletion of TβRI in combination in PTEN-inactivated endometrium [25] results in metastatic endometrial carcinoma mice.

The PI3K/AKT and TGF-β/SMAD signaling pathways are involved in the regulation of cellular processes such as cell proliferation or apoptosis. Therefore, these two signaling pathways are coordinated to integrate cellular outcomes [12]. However, the crosstalk between these two pathways is still under active investigation, and several cell type-specific mechanisms have been reported [12]. The first mechanism involves an interaction of AKT with SMAD3 in the cytoplasm, preventing its nuclear translocation and the transcriptional activation of SMAD3 target genes [26,27]. In the second proposed mechanism, AKT phosphorylates the forkhead transcription factor (FOXO) which causes its nuclear export and interferes with the formation of a transcriptionally active FOXO/SMAD transcriptional complex [28]. The third mechanism describes a collaborative effect of TGF-β/SMAD signaling loss and PI3K/AKT activation in tumor development. In this mechanism, PTEN loss and SMAD4 inactivation or inhibition through either genetic deletion of SMAD4 [29,30] leads to tumor progression in a mouse model of prostatic cancer.

Here, we provide in vivo and in vitro evidence for a regulation of SMAD2/3 by the PI3K/AKT signaling pathway. We demonstrate that SMAD2/3 is constitutively located in the nucleus of PTEN-inactivated endometrium. In the nucleus, SMAD2/3 acts as a tumor suppressor, restraining the increase of cell proliferation caused by PTEN deficiency. Moreover, we demonstrate that nuclear localization of SMAD2/3 is AKT-dependent, as its inhibition restores cytosolic localization of SMAD2/3.

## 2. Methods

### 2.1. Reagents and Antibodies

Epidermal growth factor (EGF) and LY294002 were from Sigma-Aldrich (St. Louis, MO, USA), and Matrigel^®^ (rBM) was purchased from BD Biosciences (San Jose, CA, USA). Recombinant TGF-β and Insulin−Transferrin−Sodium Selenite (ITS) supplements were from Invitrogen (Carlsbad, CA, USA). (Z)-4-Hydroxytamoxifen (TAM), BisBenzimide H 33,342 trihydrochloride (Hoechst), rhodamine conjugated-phalloidin and the TGF-β superfamily type I receptor inhibitor SB431542 were from Sigma-Aldrich (St. Louis, MO, USA). Rapamycin and Everolimus were from Selleckchem. Antibodies against TGFβRI (#sc:398), TGFβRII (#sc:220 and #sc:400), Cyclin D1 (#sc:20044), Histone H1 (#sc:8030), Pan-AKT (#sc:1618) and SMAD4 (#sc:7966) were from Santa Cruz Biotechnology (Santa Cruz, CA, USA). Antibody to α-Tubulin (#T9026) was obtained from Sigma-Aldrich (St. Louis, MO, USA); anti-antibodies to cleaved-caspase 3 (#9961), PTEN (#9188), p-AKT (ser473) (#4060) and p-SMAD2 (Ser465/467)/SMAD3 (Ser423/425) (#8828) were from Cell Signaling Technology (Beverly, MA, USA). LDH (Lactate Dehydrogenase) (#100-1173) was from Rockland Immunochemicals INC. (Limerick, PA, USA); anti-total SMAD2/3 (#610051) was purchased from BD Biosciences (San Jose, CA, USA).

PTEN cDNA from PKR5-PTEN plasmid (obtained from Rafa Pulido’s laboratory) was subcloned in the PTEN-encoding lentiviral vector. CA-AKT was a gift from Elisabeth Krizman and Michael Robinson’s laboratory.

### 2.2. Genetically Modified Mouse Models

PTEN and SMAD2 conditional knock-out mice and mice expressing Cre:ER^T^ were housed, bred and genotyped as previously described [31]. Floxed homozygous PTEN (C;129S4-*Ptentm1Hwu*/J, hereafter called PTEN^fl/fl^) Cre:ER (B6.Cg-Tg(CAG-CRE/Esr1* 5Amc/J) mice were obtained from the Jackson Laboratory (Bar Harbor, ME, USA). Cre:ER^+/−^ PTEN^fl/fl^ mice were bred in a mixed background (C57BL6; 129S4) by crossing PTEN^fl/fl^ and Cre:ER^+/−^ mice. To obtain mice carrying both PTEN floxed alleles (PTENfl/fl) and a single Cre:ER (Cre:ER^+/−^), Cre:ER^+/−^ PTEN^fl/+^ were backcrossed with PTEN^fl/fl^ mice. SMAD^fl/fl^ mice were provided by Dr. Martin M Matzuk (Department of Pathology, Baylor College of Medicine, One Baylor Plaza, Houston, TX, USA). SMAD3^fl/fl^ mice genotyping PCR was carried out with the following primer: forward primer 5′-CTC CAG ATC GTG GGC ATA CAG C-3′; SMADd3^fl/fl^ reverse primer 5′-GGT CAC AGG GTC CTC TGT GCC-3′.

To induce deletion of floxed alleles, Tamoxifen (Sigma-Aldrich T5648, St. Louis, MO, USA) was dissolved in 100% ethanol at 100 mg/mL. Tamoxifen solution was emulsified in corn oil (Sigma-Aldrich C8267) at 10 mg/mL by vortexing. To induce PTEN deletion, adult mice (4–5 weeks old) were given a single intraperitoneal injection of 0.5 mg of tamoxifen emulsion (30–35 μg per mg body weight).

### 2.3. Isolation of Endometrial Epithelial Cells and Organoid Culture

Cell culture experiments were performed in the Cell Culture Scientific and Technical Service from Universitat de Lleida, Lleida, Catalonia, Spain. Isolation and culture of endometrial organoids was performed as previously described with minor modifications [31,32]. Mice were killed by cervical dislocation and uterine horns were dissected, washed with HBSS and chopped in 3–4 mm length fragments. Uterine fragments were digested with 1% trypsin (Invitrogen) in HBSS (Invitrogen) for 1 h at 4 °C and 45 min at room temperature and epithelial sheets were squeezed-out of the uterine pieces. Epithelial sheets were washed twice with PBS and resuspended in 1 mL of DMEM/F12 (Invitrogen) supplemented with 1 mM HEPES (Sigma), 1% of penicillin/streptomycin (Sigma) and fungizone (Invitrogen) (basal medium). Epithelial sheets were disrupted mechanically in basal medium, and cells were diluted in basal medium containing 2% of dextran-coated charcoal-stripped serum (Invitrogen) and plated into culture dishes (BD Falcon). Cells were cultured for 24 h in an incubator at 37 °C with 5% CO_2_ and saturating humidity. Twenty-four hours later, cells were washed with HBSS and incubated with trypsin/EDTA solution (Sigma) for 5 min at 37 °C. Cells were collected resuspended in HBSS, and mechanically disrupted cells were centrifuged and plated in matrigrel-coated tissue culture plates in basal medium containing 3% of matrigel. Twenty-four hours after plating, the medium was replaced by basal medium supplemented with 5 ng/mL EGF and 1/100 dilution of Insulin−Transferrin−Sodium Selenite (ITS) Supplement (Invitrogen) and 3% of fresh matrigel. Medium was replaced every 2–3 days.

### 2.4. Viral Production, Infection and In Vitro Transfection Conditions

Production of lentiviruses carrying PTEN cDNA (FCIV-PTEN) or constructively active AKT (CA-AKT) was achieved by transfecting HEK293T packaging cells with linear PEI (40 µM) in combination with lentiviral plasmids and [psPAX2 packaging and pMD2G envelope] helper plasmids at 1:1:1 ratio, respectively. Viral production was performed in HEK293T cells as previously described [31].

### 2.5. Chromatin Immunoprecipitation (ChIP)

ChIP analysis was performed as previously described [31]. Endometrial organoids were crosslinked at room temperature for 15 min by adding 1/10 volumes of 11% formaldehyde solution and quenched with glycine for 15 min. Organoids were washed in ice-cold PBS and lysed. Lysate was sonicated ten times with the following protocol: 4 cycles of 30″ sonication and 20″ pause at 20% amplitude. Ten µL were separated as a whole cell extract input. Anti-SMAD2/3 conjugated beads were collected and washed with blocking buffer. 100 µL of nuclear extracts were added to the bead solution and incubated overnight at 4 °C. The next day, beads were consecutively washed with low and high salt buffer, LiCl wash buffer and with TE buffer and resuspended with elution buffer for 15 min at 65 °C. Elution buffer was added to input. Input and pellet were incubated in the oven overnight at 65 °C. Lysates were centrifuged and supernatants were digested and incubated at 37 °C for 2 h in TE buffer containing RNaseA and proteinase K and incubated for 2 h at 55 °C. DNA was extracted by phenol/chloroform/isoamyl alcohol method and precipitated. Samples were incubated at −20 °C for 4 h, and then the samples were centrifuged at top speed. The pellets were washed with 80% EtOH. Samples were spun again, and the pellet was resuspended in TE buffer and incubated at 65 °C on a heating block for 15 min. Primers used for PCR were: PTEN PROMOTER −1/−500 (F TCGGAAAGCCGGAGGGGAG, R GTGTCTCCCGCGTGGGTCA); PTEN PROMOTER −501/−986 (F TGACCCACGCGGGAGACAC, R GGCCTGGGAGGGCTCAAAG); PTEN PROMOTER −987/−1459 (F CTTTGAGCCCTCCCAGGCC, R CAACCGTGGGAGAAGAGGC).

### 2.6. Western Blotting

Western blotting was performed as previously described [31]. Organoids were washed with PBS and lysed with lysis buffer containing 2% SDS, 125 mmol/L Tris-HCl, pH 6.8. Protein was quantified by loading on a 10% acrylamide gel, transferring to polyvinylidene difluoride membranes (Millipore Corporation, Kenilworth, NJ, USA) and blotting with anti-tubulin antibody. Band density was determined by using Image Lab 4.0.1 software (Bio-Rad laboratories, Richmond, CA, USA). Equal amounts of proteins were subjected to SDS-polyacrylamide gel electrophoresis and transferred to polyvinylidene difluoride membranes. Nonspecific binding was blocked by incubation with TBST (20 mM Tris-HCl [pH 7.4], 150 mM NaCl and 0.1% Tween 20) plus 5% of nonfat milk. Membranes were incubated with the primary antibodies overnight at 4 °C and for 1 h room temperature with secondary horseradish peroxidase (1:10,000 in TBST). Signal was detected with ECL Advance (Amersham-Pharmacia, Little Chalfont, Buskinghamshire, UK) and SuperSignal West Femto Trial Kit (Thermo Scientific, Rockford, IL, USA).

### 2.7. Human Tissue Samples Selection and Tissue Micro Arrays (TMAs) Construction

Three TMAs were constructed using the manual arrays from Beecher Instruments^TM^. The TMAs contained formalin-fixed, paraffin-embedded (FFPE) tissue from 79 primary Endometrioid Endometrial Carcinomas (EEC). The tumors were classified following the most recent WHO criteria. They were surgically staged and graded according to the International Federation of Gynecology and Obstetrics (FIGO) grading systems. They included 19 grade 1 EECs, 23 grade 2 EECs and 37 grade 3 EECs. Samples were obtained from the surgical pathology specimens. The study complied with Law 14/2007 and RD 1716/2011 of the Autonomous Community (Generalitat of Catalonia), Spanish Government and EU Directives and was approved by the Ethics Committee of Hospital Arnau de Vilanova de Lleida (CEIC). Informed consent was obtained from each patient. All tissue samples were histologically reviewed by two members of the team, and representative tumor or non-tumor areas were marked in the corresponding paraffin blocks. Tissue cylinders with a diameter of 0.6-mm were punched from two different tumor areas of each “donor” tissue block and brought into a recipient paraffin block.

### 2.8. Total RNA Extraction, Reverse Transcriptase−Polymerase Chain Reaction (RT-PCR) and Quantitative Real-Time PCR RT-qPCR

Total RNA was extracted from the uterine endometrium using the RNeasy Total RNA kit (Qiagen, Valencia, CA, USA). For RT-qPCR assays, cDNA was amplified by heating at 95 °C for 10 min, followed by 40 PCR cycles of 95 °C for 15 s and 60 °C for 1 min using the ABI Prism 7900 Sequence Detection System (Applied Biosystems) and Promega GoTaq^®^ qPCR Master Mix (Madison, WI, USA). Relative mRNA expression levels were calculated by using the 2ΔΔCt method and are presented as ratios to the housekeeping gene glyceraldehyde-3-phosphate dehydrogenase (GAPDH). Taqman^®^ technology from Applied Biosystems was used for RT-qPCR analyses. The probes were: GAPDH, Mm99999915_g1; SMAD2, Mm00487530_m1; SMAD3, Mm01170760_m1. The number of cycles required to reach the crossing point for each sample was used to calculate the amount of each product using the 2-CP method. Each sample pool was amplified in triplicate using GAPDH for normalization.

### 2.9. Immunohistochemistry

Mice uteri were dissected, washed with PBS, fixed in 10% neutral-buffered formalin, embedded in paraffin and sectioned (4–5 μm). Mice uteri and TMA blocks from human tissue samples were sectioned at a thickness of 3 μm, dried, rehydrated and submitted to antigen retrieval for 20 min in 50× Tris/EDTA buffer, pH 9 in the Pre-Treatment Module, PT-LINK (DAKO) at 95 °C. Endogenous peroxidase was blocked. The antibodies used were against TGFβ1, TGFβRII, SMAD 2/3, SMAD4 and PTEN (6H2.1). The reaction was visualized with the EnVisionTM FLEX Detection Kit (DAKO, Glostrup, Denmark) for SMAD 2/3, SMAD 4 and PTEN and EnVisionTM FLEX+ rabbit (LINKER) Detection Kit (DAKO, Glostrup, Denmark) using diaminobenzidine chromogen as a substrate. Sections were counterstained with hematoxylin. Appropriate negative controls including no primary antibody were also tested.

Immunohistochemical results shown in Appendix A were evaluated by following uniform pre-established criteria. Immunostaining was graded semi-quantitatively by considering the percentage and intensity of the staining. A histological score was obtained from each sample and values ranged from 0 (no immunoreaction) to 300 (maximum immunoreactivity). The score was obtained by applying the following formula: Histoscore = 1 × (% light staining) + 2 × (% moderate staining) + 3 × (% strong staining). The histological score was also used for evaluation of cytosolic and nuclear staining intensity.

In the case of TMA evaluation, immunohistochemical evaluation was done after examining the two different tumor cylinders from each case. PTEN immunoreactivity was scored as follows: 2 for highly expressing cylinders, 1 for moderately expressing cylinders and 0 for cylinders completely lacking PTEN expression. For evaluation of SMAD2/3 for cytosolic and nuclear staining intensity, cylinders were scored as follows: *n* > c for cylinders showing only nuclear expression; *n* < c for cylinders showing only cytoplasmic expression; *n* = c for cylinders showing both nuclear and cytosolic expression. The reliability of such scores for interpretation of immunohistochemical staining in EC TMAs has been shown previously [33,34].

To support the scoring of immunohistochemistry, an automated imaging system, the ACIS^®^ III Instrument (DAKO, Glostrup, Denmark), was also used. An intensity score, which ranged from 60 to 255, was obtained from 4 different areas of each sample.

### 2.10. Immunofluorescence Study

Immunohistochemical and immunofluorescence experiments were performed as previously described [31]. Organoids were fixed for 5 min at room temperature with formalin and washed with PBS. Depending on primary antibody, cells were permeabilized with 0.2% Triton (T) X-100 in PBS for 10 min or with 100% methanol (Me) for 2 min. Organoids were incubated overnight at 4 °C with the indicated dilutions of antibodies: SMAD2/3 (T), TGFβRI (T), TGFβRII (T), α-Tubulin (T) and anti-SMAD4 (Me), washed with PBS and incubated with Alexa Fluor secondary anti-mouse or anti-rabbit antibodies (1:500) containing 5 μg/mL of Hoechst 33,342 in PBS at room temperature for 4 h. For double-immunofluorescence, organoids were incubated with the second round of primary and secondary antibodies. For all double-immunofluorescence stains, first and second primary antibodies were from a different isotype. Immunofluorescence staining was visualized and analyzed using confocal microscopy (model FV1000; Olympus, Tokyo, Japan) with the 10× and the oil-immersion 60× magnification objectives. Analysis of images was obtained with Fluoview FV100 software (Olympus, Shinjuku City, Tokyo, Japan).

### 2.11. Confocal Imaging and Evaluation of SMAD2/3 Positive Nuclei and Glandular Perimeter Measurement

Images of endometrial epithelial spheroids were captured and digitized with a confocal microscope (Fluoview FV1000-Olympus). Epithelial perimeter analysis was processed by image analysis software (ImageJ version 1.46r; NIH, Bethesda, MD, USA), generating binary images of the spheroids as previously described. For each experiment, at least 150 spheroids were quantified. SMAD2/3 nuclei were scored and divided by the total number of cells (visualized by Hoechst staining). The results are expressed as a percentage of SMAD2/3-positive nuclei cells. The investigators were not blinded to allocation during experiments or outcome assessment.

### 2.12. Statistical Analysis

TMA statistical analyses were performed using linear mixed models to assess the effects of any experimental factor on PTEN staining. For each experimental design, SEs were used to statistically assess the main effect of each variable but also their paired interactions. Chi-squared test was conducted to assess the reduction in the levels of PTEN expression (considered categorically as 0, 1 or 2) in relation to SMAD2/3 expression and whether the expression was higher in the nucleus versus the cytoplasm. This analysis was performed globally for all EEC cases and separately for grades I, II or III. Values are presented in the graphs as the mean ± standard errors of the mean (SEM) of *n* cells cultures experiments or *n* biopsies where each value is the average of responses in triplicate, at least.

The normality of the distribution of experiments was assessed by Kolmogorov−Smirnov test. No statistical method was used to predetermine sample size. Statistical analysis was performed with GraphPad Prism 8.0. Differences between two groups were assessed by Student’s *t* test (unpaired or paired as needed depending on the study design). Differences between more than two groups were assessed by one-way ANOVA, followed by Tukey’s multiple comparison test or two-way ANOVA, followed by the Bonferroni post hoc comparison test. A *p* < 0.05 was considered statistically significant. All data examined are expressed as mean ± SEM.

## 3. Results

### 3.1. PTEN Deficiency Leads to Constitutive Phosphorylation and Nuclear Localization of SMAD2/3

The effects of the PI3K/AKT signaling pathway on SMAD signaling and its downstream effects are still controversial, and opposing effects of PI3K/AKT signaling have been reported. Therefore, we intended to investigate the effects of PTEN deficiency on TGF-β/SMAD signaling on endometrial cells. For this purpose, PTEN Cre:ER^+/−^;PTEN^fl/fl^ organoids treated or not with tamoxifen to induce PTEN excision were collected and split in two fractions. The first fraction was used for total protein extraction and subjected to analysis by Western blot. The second fraction was used for cytosolic and nuclear protein extraction. Western blot analysis of total protein cell lysates revealed that PTEN deficiency results in a marked increase of SMAD2/3 expression and phosphorylation (Figure 1A and Appendix A). To ascertain whether the increase of SMAD2/3 expression was due to an increase of its expression, we performed an RT-PCR analysis of SMAD2, SMAD3 and SMAD4 mRNA of Cre:ER^+/−^;PTEN^fl/fl^ treated with tamoxifen to induce PTEN deficiency. The expression of SMADs was not modified by PTEN deletion, indicating that the increase of protein appears to be caused by post-transcriptional mechanisms (Figure 1B). To further analyze subcellular distribution of SMADs after PTEN deletion, Cre:ER^+/−^;PTEN^fl/fl^ organoid cultures were treated with tamoxifen to induce PTEN deletion, and SMAD2/3 and SMAD4 localization was assessed by immunofluorescence. In PTEN wild-type glandular structures (Cre:ER^+/−^;PTEN^fl/fl^ glands without induction of PTEN deletion), addition of TGF-β triggered a rapid nuclear translocation of SMAD2/3. In contrast, in PTEN-deficient cultures, SMAD2/3 was constitutively located in the nucleus, even in the absence of TGF-β stimulation (Figure 1C). After 6 h of TGF-β treatment, PTEN wild-type cells showed SMAD2/3 localization back to the cytoplasm. In contrast, in PTEN-deficient cells, SMAD2/3 was retained in the nucleus (Figure 1C and Appendix A). Immunofluorescence of SMAD4 showed its nuclear translocation upon TGF-β treatment in both wild-type cells and PTEN knock-out cells; surprisingly, however, SMAD4 was retained in the cytoplasm of non-stimulated PTEN-deficient cells (Figure 1D and Appendix A). To further evidence such differential localization of SMAD2/3 and SMAD4 in PTEN-deficient cells, we performed nuclear and cytosolic fractionation of PTEN Cre:ER^+/−^;PTEN^fl/fl^ organoids treated or not with tamoxifen to induce PTEN excision. Western blot analysis of SMAD2/3 and SMAD4 distribution further demonstrated that PTEN loss leads to a selective SMAD2/3, but not SMAD4 nuclear translocation (Figure 1E).

Having demonstrated that PTEN deficiency results in a constitutive nuclear localization of SMAD2/3 in organoids, we aimed to study SMAD2/3 localization in PTEN-deficient endometrium in vivo. For this purpose, we carried out immunohistochemistry to localize SMAD2/3 on uterine sections obtained from Cre:ER^+/−^;PTEN^fl/fl^ mice injected or not with tamoxifen. It has been previously described that administration of a single dose of tamoxifen to Cre:ER^+/−^;PTEN^fl/fl^ mice results in a mosaic of PTEN deletion in the endometrium [35]. In this mouse model, PTEN-negative tumoral endometrial epithelium co-exists with normal endometrial epithelial cells that retain PTEN expression. This mouse model allows the study of SMAD2/3 expression in PTEN-deficient and PTEN wild-type cells in the same uterine section of a single mouse. Endometrial glands displaying negative PTEN immunostaining showed nuclear expression of SMAD2/3, whereas glands retaining PTEN expression displayed more cytoplasmic staining (Figure 2A). As we observed in the Western blot analysis of SMAD2/3 in PTEN-deficient organoids (Figure 1A), immunohistochemical analysis also evidenced a significant increase of global SMAD2/3 staining in tissues lacking PTEN expression. The increase of nuclear SMAD2/3 in PTEN-deficient glands was further validated using tamoxifen-treated and non-treated littermates (Appendix A). To rule out the possibility that PTEN was influencing the expression of other TGF-β signaling components, we also performed immunohistochemical analysis of SMAD4 and TβRII in serial sections of endometrial tissue. SMAD4 and TβRII showed no differences on their expression or localization between PTEN-positive or PTEN-negative glands (Figure 2A).

One of our main concerns of our results was the specificity of SMAD2/3 immunostaining. To demonstrate the specificity of SMAD2/3 nuclear staining in PTEN-deficient cells, we performed an immunofluorescence on organoid culture obtained from Cre^+/−^; Smad2^fl/fl^; Smad3^fl/fl^ in which we induced SMAD2/3 ablation by tamoxifen treatment. Tamoxifen-induced deletion of SMAD2/3 caused a complete lack of labeling with the antibody used throughout our study (Appendix A). This result rules out the possibility that nuclear translocation of SMAD2/3 observed in immunostaining is due to unspecific antibody labeling.

Finally, we sought to investigate whether PTEN deficiency led to nuclear localization of SMAD2/3 in human endometrial carcinomas. To detect and study the association between SMAD2/3 localization and PTEN expression, we performed immunohistochemical analysis on EEC samples from human tissue. Interestingly, grade III EECs but not grade I and grade II EECs displaying decreased PTEN expression were associated with a significant increase of nuclear SMAD2/3 staining (*p* = 0.02, Figure 2B).

### 3.2. Nuclear Translocation of SMAD2/3 Is Independent of TGF-β Receptor Activation

Next, we investigated the molecular mechanism by which PTEN deficiency could cause nuclear translocation of SMAD2/3. The regulation of SMAD2/3 activity and localization by PI3K/AKT signaling is not fully understood, and different mechanisms have been proposed [12]. Among them, it has been reported that AKT signaling can promote TβRs delivery to the cell surface, resulting in an enhanced autocrine TGF-β signaling and therefore increased SMAD3 nuclear translocation [36]. To test whether such mechanism may explain the constitutive nuclear localization of SMAD2/3 downstream of PTEN ablation, we analyzed the localization of SMAD2/3 by immunofluorescence on PTEN wild-type and PTEN-deficient 3D cultures treated with the TβR inhibitor SB431542. The addition of SB431542 failed to restore cytosolic localization of SMAD2/3 in PTEN-deficient cells, suggesting that TβRs activation is not involved in translocation of SMAD2/3 after PTEN deletion (Figure 3A and Appendix A). These results were further confirmed by ChiP analysis of SMAD2/3 binding to PTEN promoter. The addition of SB431542 completely blocked TGF-β-induced SMAD2/3 binding to PTEN promoter, but it was unable to reverse constitutive binding of SMAD2/3 to PTEN promoter in PTEN-deficient cells (Figure 3B). Furthermore, analysis of TGFβRI and TGFβRII on uterine sections from Cre:ER^+/−^;PTEN^fl/fl^ mice injected with tamoxifen revealed no differences in their pattern of expression between PTEN wild-type and PTEN-deficient tissue in vivo (Figure 3C). Consistently, immunofluorescence analysis of TGFβRI and TGFβRII expression on 3D cultures Cre:ER^+/−^;PTEN^fl/fl^, treated or not with tamoxifen to induce PTEN deletion, showed no differences in their expression patterns (Figure 3D).

### 3.3. Activation of PI3K/AKT/mTOR Signaling Axis Triggers Nuclear Localization of SMAD2/3 Downstream of PTEN Deletion

To further investigate the molecular link between PTEN deletion and SMAD2/3 translocation, we dissected the PI3K/AKT signaling pathway. First, to directly address whether PTEN deficiency was responsible for the constitutive translocation of SMAD2/3 to the nucleus, PTEN expression was restored in PTEN knock-out organoid cultures. For this purpose, PTEN-deficient cultures were infected with lentiviruses encoding PTEN cDNA. Exogenous PTEN expression caused an increase of SMAD2/3 localization into the cytoplasm (Figure 4A). Next, to demonstrate that SMAD3 nuclear translocation was caused by enhanced PI3K/AKT activity in PTEN-deficient cells, organoids were treated with TGF-β plus the PI3K inhibitor LY294002, and SMAD2/3 localization was assessed by immunofluorescence. As a control, matched organoid cultures from Cre:ER^+/−^;PTEN^fl/fl^ were analyzed by Western blot to check that the addition of LY294002 was able to reduce Akt phosphorylation (Appendix A). In wild-type organoid cultures (Cre:ER^+/−^;PTEN^fl/fl^ without tamoxifen treatment), TGF-β induced a rapid translocation of SMAD2/3, but interestingly, such translocation was not inhibited by the addition of LY294002. These results suggested that TGF-β-induced translocation of SMAD2/3 in PTEN-proficient cells was independent of PI3K activity (Figure 4B). In contrast, Cre:ER^+/−^;PTEN^fl/fl^ treated with tamoxifen (PTEN-deficient cells) displayed nuclear SMAD2/3 staining, but in this case, the inhibition of PI3K/AKT by LY294002 restored cytosolic distribution of SMAD2/3 (Figure 4B and Appendix A). It is noteworthy that LY294002 had no effect on nuclear translocation of SMAD2/3 in PTEN-deficient cells stimulated with TGF-β. These results suggest nuclear translocation of SMAD2/3 caused by PTEN deletion is dependent on PI3K activity.

Besides AKT, PI3K activates other downstream kinases [37]. To demonstrate that AKT was the kinase responsible for SMAD3 nuclear localization, we infected organoids with lentiviruses carrying a constitutively active form of AKT (CA-AKT) and we analyzed SMAD2/3 cellular localization by immunofluorescence. Notably, CA-AKT expression caused a marked increase of cells displaying nuclear localization of SMAD3 (Figure 4C), suggesting that AKT activity plays a pivotal role in the regulation of SMAD2/3 localization.

Once AKT is activated, it also phosphorylates many downstream substrates [38]. Among them, mTOR complex 1 (mTORC1) is one of the downstream AKT substrates [39] that has been shown to play a pivotal role in the development and progression of PTEN-deficient endometrial tumors. For this reason, we questioned whether mTORC1 could be the molecular target of AKT to mediate SMAD2/3 nuclear localization. To address this hypothesis, we treated PTEN-deficient organoids with two mTOR inhibitors, Rapamycin and Everolimus. SMAD2/3 immunofluorescence revealed that both inhibitors completely restored cytosolic localization of SMAD2/3 (Figure 4D and Appendix A). Furthermore, both inhibitors also prevented nuclear translocation of SMAD2/3 in organoids infected with CA-AKT (Figure 4E).

### 3.4. Nuclear SMAD2/3 Render Organoid Cultures Resistant to TGF-β-Induced Apoptosis and Restrains Cell Proliferation in PTEN-Deficient Cells

Finally, we investigated whether nuclear translocation of SMAD2/3 caused by PTEN had a tumor-suppressive or tumor-promoting function. To assess this question, we analyzed the effects of SMAD2/3 knock-out on PTEN-deficient endometrial cells. For this purpose, organoid cultures from Cre:ER^+/−^;PTEN^fl/fl^;SMAD2^+/+^;SMAD3^+/+^ (PTEN knock-out, PTEN KO), Cre:ER^+/−^;PTEN^+/+^;SMAD2^fl/fl^;SMAD3^fl/fl^ (SMAD2/3 double knock-out, S2/3 dKO) and Cre:ER^+/−^;PTEN^fl/fl^;SMAD2^fl/fl^l;SMAD3^fl/fl^ (PTEN and SMAD2/3 triple knock-out, PTEN;S2/3 tKO) were treated with tamoxifen to induce excision of floxed alleles. As a control, we used Cre:ER^+/−^;PTEN^fl/fl^;SMAD2^fl/fl^;SMAD3^fl/fl^ without tamoxifen addition (WT). As we have previously demonstrated [31], PTEN KO displayed a marked increase in glandular perimeter compared to WT organoids. Similarly, S2/3 dKO also showed an increase in glandular perimeter (Figure 5A,B) suggesting that SMAD2/3 also acts as tumor suppressor. PTEN;S2/3 tKO organoids displayed a significant further increase of glandular perimeter over PTEN KO or S2/3 dKO (Figure 5A,B), indicating that SMAD2/3 restrain cell growth in PTEN-deficient cells.

Previous results from our laboratory have also demonstrated that either PTEN or SMAD3 deficiency results in the blockade of TGF-β-induced apoptosis. Here, we also analyzed the effects of TGF-β stimulation on WT, SMAD2/3 dKO and PTEN;S2/3 tKO organoids. Cleaved caspase-3 immunostaining revealed that organoid cultures deficient for PTEN and SMAD2/3 were completely resistant to TGF-β-induced apoptosis (Figure 5C). These results further support the tumor-suppressive function of SMAD2/3 in a PTEN-deficient context.

## 4. Discussion

TGF-β is a cytokine that regulates a myriad of cellular functions depending on cell type and context [1]. During cancer development, TGF-β is a tumor suppressor on normal or pre-malignant cells, but it is a potent tumor promoter in malignant stages. However, the molecular mechanisms of such opposing effects are not fully understood. It is widely accepted that in epithelial tissues, TGF-β tumor-suppressive action depends on its ability to induce cell growth arrest or apoptosis [9]. In the uterus, TGF-β plays an important role in endometrial development and physiology [20] and also in endometrial carcinogenesis. TGF-β signaling is impaired during endometrial carcinogenesis [40], and TGF-β signaling downregulation has been associated with poor prognosis [41]. In addition, during early stages of endometrial carcinogenesis, impaired TGF-β signaling correlates with loss of growth inhibition [42]. On the contrary, TGF-β promotes epithelial-to-mesenchymal transition (EMT) and increases invasiveness of endometrial cancer cell lines [43]. In recent years, the role of TGF-β/SMAD in endometrial carcinogenesis signaling has been uncovered by conditional deletion of TGF-β receptors and SMADs: uterine conditional deletion of TβRI [23], conditional double deletion of SMAD2 and SMAD3 [24] or conditional deletion of TβRI in combination in PTEN-inactivated endometrium [25] results in metastatic endometrial carcinoma mice.

Here, we have demonstrated that lack of PTEN results in an increase in PI3K/AKT signaling which leads to constitutive nuclear translocation of SMAD2/3. The PI3K/AKT signaling pathway plays a pivotal role in the regulation of endometrial homeostasis. Perturbations of this signaling pathway are the most frequent molecular alterations found in endometrial cancers [14,15,16]. Activation of the PI3K/AKT signaling pathway leads to survival and proliferation, and therefore, it can abrogate pro-apoptotic or cytostatic effects of TGF-β. Mechanistically, the regulation of SMADs activation and their nuclear translocation by the PI3K/AKT signaling pathway is still controversial, and opposing effects of PI3K/AKT activation on SMAD activity and localization have been observed. On the one hand, it has been reported that AKT can directly interact with SMAD3 inhibiting its nuclear translocation and activation [26,27]. Moreover, activation of PI3K/AKT signaling by IGF-1 suppresses SMAD3 activation in prostate cells [44]. On the other hand, it has been also demonstrated that enhanced PI3K/AKT signaling triggers SMAD activation in several cell types with different cellular outcomes. In keratinocytes, loss of PTEN increases TGFβ-mediated invasion with enhanced SMAD3 transcriptional activity [45]. In the kidney, PTEN loss initiates tubular dysfunction via SMAD3-dependent fibrotic responses [46]. Prostates from PTEN-deficient mice display increased phosphorylation and activation of SMAD3 and SMAD4 [29]. We have also addressed the molecular mechanism by which loss of PTEN causes nuclear translocation of SMAD2/3. It has been reported that PI3K/AKT activation increases TGF-β receptors in the cell surface, resulting in an enhanced autocrine TGF-β signaling that causes SMAD3 activation [36]. SMAD2/3 activation downstream PTEN deletion is dependent of PI3K/AKT signaling but independent of TGF-β receptors. In contrast, we have unveiled the PI3K/AKT/mTORC1 signaling pathway as the major one responsible for SMAD2/3 nuclear translocation in PTEN knock-out cells. It is worth highlighting that SMAD2/3 translocation can be blocked by mTORC1 inhibitors such as Everolimus, which is a therapeutic agent for PTEN-deficient cancers [47]. At the functional level, mTORC1 inhibition restores TGF-β-induced apoptosis downstream of PTEN loss or constitutive AKT activation. Therefore, apart from new mechanistic insight on the regulation of SMAD2/3 by PTEN, or findings could have a therapeutic value. Finally, we would like to highlight that the mechanistic differences between our model and others can be explained by the well-known cell type or cell context specificity of TGF-β signaling [1].

Another observation that deserves discussion is the role of SMAD4 to drive TGF-β-induced cellular responses. Most of the cell responses activated by TGF-β require association of R-SMADs (SMAD2/3) with SMAD4. However, an increasing number of evidences demonstrate that SMAD2 and SMAD3 may have different functions in TGF-β signaling [48], independently of SMAD4. To this end, our results demonstrate PTEN deficiency caused constitutive nuclear translocation of SMAD2/3, while SMAD4 was still retained in the cytoplasm.

Besides the results derived from organoid cultures, one of the strengths of our findings is the nuclear localization of SMAD2/3 in both mouse and human PTEN-deficient endometrial samples in vivo. Our mouse model of tamoxifen-induced PTEN deletion is a mosaic where cells lacking PTEN that develop endometrial tumors are nearby cells keeping PTEN expression that show normal phenotype. It is noteworthy that all PTEN-deficient cells display nuclear translocation of SMAD2/3, whereas in the same sample, cells retaining PTEN expression do not have nuclear staining for SMAD2/3. More importantly, nuclear SMAD2/3 in PTEN-deficient mouse endometrial cancer is extensible to human endometrium. The analysis of human endometrial carcinomas revealed a significant inverse correlation between PTEN expression and SMAD2/3 nuclear staining in Grade III EC. It is worth mentioning this and considering it as high-risk EC that often spreads to other parts of the body. This result opens the door for a further investigation of SMAD2/3 as a biomarker of PTEN deficiency in Grade III EC.

Finally, we intended to evaluate the function of SMAD2/3 in PTEN-deficient cells. It is widely accepted that in normal epithelial cells, TGF-β-induced SMAD activation is a tumor suppressor pathway, but oncogenic mutations can switch the tumor-suppressive functions of TGF-β/SMAD signaling to tumor-promoting ones. We have previously described that SMAD2/3 nuclear translocation triggered by TGF-β induces apoptosis of normal endometrial organoid cultures, and its deletion leads to increased cell proliferation and inhibition of apoptosis. Therefore, it is intriguing that PTEN deficiency, which inhibits TGF-β-induced apoptosis and leads to increased proliferation, also triggers SMAD2/3 nuclear translocation. This observation raises the question: is the nuclear translocation of SMAD2/3 downstream of PTEN deletion, a tumor suppressor or tumor-promoting mechanism? By concomitant deletion of PTEN and SMAD2/3, we have demonstrated that SMAD2/3 restrains tumor proliferation of PTEN-deficient organoids. PTEN loss leads to an increase of organoid size and triggers SMAD2/3 nuclear translocation. When PTEN and SMAD2/3 and organoid size are further increased, this indicates that SMAD2/3 nuclear translocation constrains tumorigenesis triggered by PTEN loss. These results strongly suggest that SMAD2/3 has tumor-suppressive functions when it is translocated to the nucleus by PTEN deficiency.

It is well known that SMAD signaling is capable of triggering EMT in initiated or malignant cells. However, neither SMAD2/3-deficiency nor SMAD2/3-PTEN triple deficiency caused any sign of morphological changes compatible with EMT. Therefore, the tumor-suppressive function of SMADs is restricted to the control of glandular proliferation and the regulation of apoptosis triggered by TGF-β.

## 5. Conclusions

To conclude, in the present study, we have investigated the effects of PTEN deficiency in TGF-β/SMAD signaling and its role in endometrial carcinogenesis. We demonstrate that lack of PTEN triggers a PI3K/AKT-dependent nuclear localization of SMAD2/3, which acts as a tumor suppressor that restrains proliferation endometrial cells lacking PTEN. Moreover, the strong correlation between PTEN deficiency and SMAD2/3 raises the possibility that SMAD2/3 can be used as a biomarker of grade III EECs harboring PTEN alterations.

## Figures and Tables

**Figure 1 cancers-13-04990-f001:**
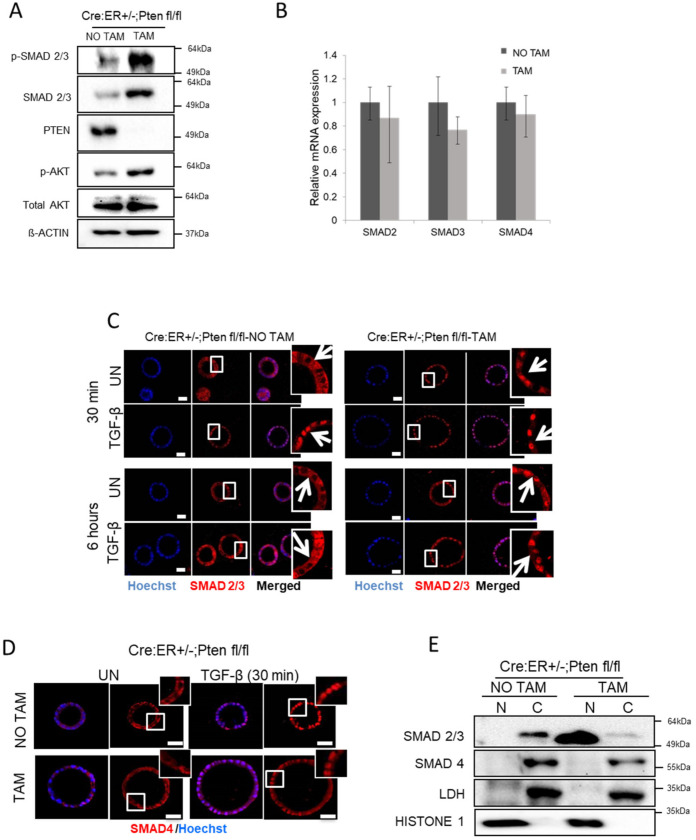
PTEN deficiency triggers constitutive nuclear localization of SMAD2/3 in endometrial organoids. (**A**) Western blot analysis of p-SMAD2/3, SMAD2/3 and p-AKT on lysates from Cre:ER^+/−^;PTEN ^fl/fl^ organoid cultures treated (TAM) or not (NO TAM) with tamoxifen. Membranes were also blotted with PTEN antibody to show its total deletion after tamoxifen treatment. Membranes were also reblotted with total AKT and β-actin antibodies to show equal protein loading. A representative image of *n* = 3 biological replicates is shown. (**B**) RT-qPCR analysis of SMAD2, SMAD3, SMAD4 mRNA corresponding to organoid cultures from Cre:ER^+/−^;PTEN^fl/fl^ organoid cultures treated (TAM) or not (NO TAM) with tamoxifen. (**C**) SMAD2/3 immunofluorescence on Cre:ER^+/−^;PTEN^fl/fl^ organoid endometrial organoids incubated (TAM) or not (NO TAM) with tamoxifen delete PTEN and then treated for 30 min with 10 ng/mL TGF-β, 6 h or left untreated (UN). Organoid cultures were counterstained with Hoechst to show nuclei. Magnification images of framed regions of sample are shown to show SMAD2/3 cellular localization. Arrows indicate presence or absence of nuclear staining. Scale bars: 25 μm. Data are from *n* = 3 experimental replicates (independent organoid cultures). (**D**) Representative SMAD4 immunofluorescence on Cre:ER^+/−^;PTEN^fl/fl^ endometrial cultures incubated (TAM) or not (NO TAM) with tamoxifen to induce PTEN deletion and then treated with 10 ng/mL TGF-β for 30 min or untreated (UN). Magnification images of framed regions of the samples show SMAD4 cellular localization. Data are from *n* = 3 experimental replicates (independent organoid cultures). Scale bars: 25 μm. (**E**) Western blot analysis of SMAD2/3 and SMAD4 on nuclear (N) and cytosolic (C) fractions of Cre:ER^+/−^;PTEN^fl/fl^ 3D cultures treated (TAM) or not (NO TAM) with tamoxifen. Membranes were also probed with Histone H1 and LDH to demonstrate correct nuclear and cytosolic fractionation. A representative image of *n* = 3 biological replicates is shown. Original Western blot data is shown in Appendix A.

**Figure 2 cancers-13-04990-f002:**
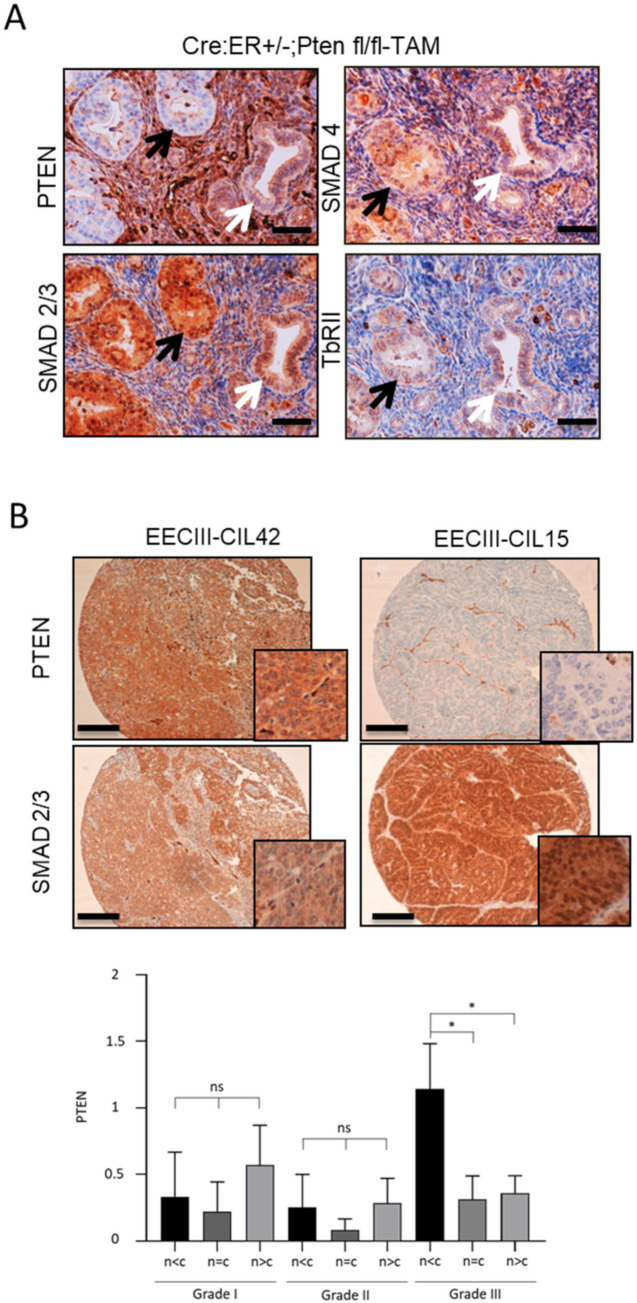
PTEN deficiency leads to nuclear constitutive localization of SMAD2/3 in vivo. (**A**) PTEN and SMAD2/3 immunohistochemistry on formalin-fixed paraffin-embedded endometrial tissue sections from 12-week-old Cre:ER^+/−^;PTEN^fl/fl^ mice that have been injected with tamoxifen (TAM) to induce PTEN deletion (4 weeks). 20× images. Black arrows indicate PTEN-negative glands that display nuclear SMAD2/3 staining. White arrows indicate glands that keep PTEN expression and display a more cytoplasmatic SMAD2/3 staining. Scale bar: 100 μm. (**B**) Representative images of PTEN and SMAD2/3 immunohistochemistry on formalin-fixed paraffin-embedded endometrial tissue sections from two human grade III endometrial carcinomas. Correlation of PTEN expression values and SMAD2/3 expression in the nucleus in grade I, grade II and grade III EECs. (*n* < c) designates more expression in the cytoplasm than in the nucleus, (*n* = c) equal expression and (*n* > c) higher expression in the nucleus than in the cytoplasm (bottom graph). Vertical bars represent ± standard error. Plot evidences a reduction of PTEN expression when the expression of SMAD2/3 in the nucleus is higher than in the cytoplasm (*p* = 0.02) in *n* = 37 EEC grade III (20× images and magnifications). Scale bar: 100 μm. * *p* < 0.05 by *t*-test analysis.

**Figure 3 cancers-13-04990-f003:**
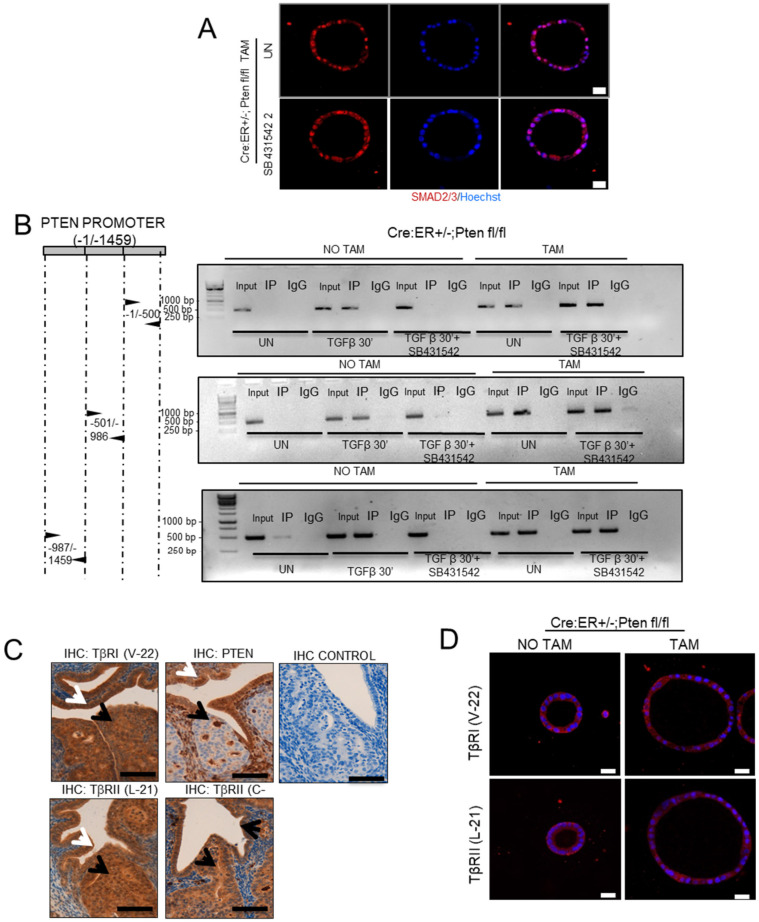
SMAD2/3 nuclear translocation is independent of TGF-β receptor. (**A**) SMAD2/3 immunofluorescence on Cre:ER^+/−^;PTEN^fl/fl^ organoids incubated with tamoxifen (TAM) to induce the deletion of PTEN and treated with SB431542 10 µM or left untreated (UN). To evidence morphology, nuclei were counterstained with Hoechst. Scale bars: 25 μm. Data are from *n* = 3 experimental replicates (independent organoids cultures). (**B**) ChIP of SMAD2/3 binding to PTEN promoter. Organoids from Cre:ER^+/−^;PTEN^fl/fl^ treated with tamoxifen (TAM) or not (NO TAM) to delete PTEN were pretreated or not with SB431542 10 µM for 2 h and then stimulated for 30 min with TGF-β. Untreated (UN) or TGF-β stimulated organoids were lysed and immunoprecipitated (IP) with control antibody (IgG) or SMAD2/3. Immunoprecipitates were subjected to PCR analysis using primers to amplify three ~500 bp segments of PTEN promoter (−1/−500, −501/−986, −987/−1459). (**C**) TbRI (antibody V-22) and TGFβRII (two different antibodies L-21 and C-16), PTEN immunohistochemistry and negative control on serial sections of endometrial tissue were obtained from 1-week-old Cre:ER^+/−^;PTEN^fl/fl^ mice; injected with tamoxifen to induce PTEN deletion. Mice were sacrificed 4 weeks after tamoxifen injection. Black arrows indicate PTEN-deficient glands in serial sections. White arrows indicate PTEN-positive gland in serial sections. (40× magnifications; *n* = 10 Cre:ER^+/−^;PTEN^fl/fl^l-TAM mice). Scale bar: 100 µM. (**D**) TGFβRI (antibody V-22, red) and TGFβRII (antibody L-21, red) immunofluorescence analysis on Cre:ER^+/−^;PTEN^fl/fl^ organoids incubated (TAM) or not (NO TAM) with tamoxifen to induce PTEN deletion. Organoids were counterstained with Hoechst (Blue) to evidence nuclei. Scale bars: 25 μm. Data are from *n* = 3 experimental replicates (independent organoid cultures).

**Figure 4 cancers-13-04990-f004:**
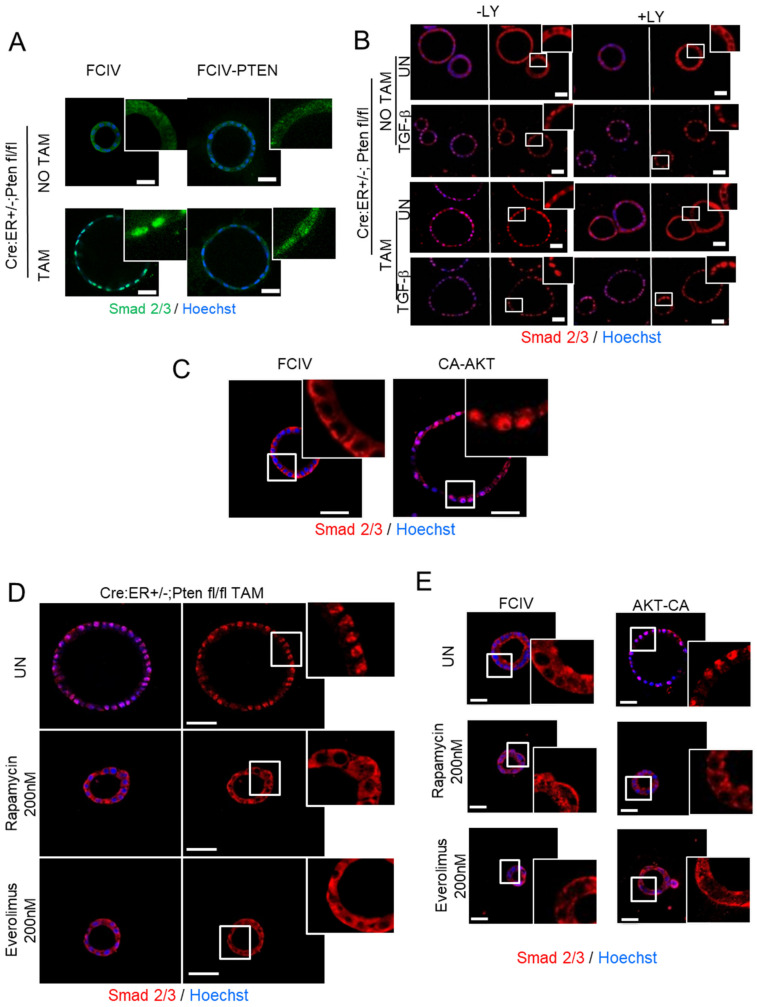
PI3K/AKT/mTORC1 inhibition restores cytosolic localization of SMAD2/3. (**A**) Representative images of SMAD2/3 immunostaining on Cre:ER^+/−^;PTEN^fl/fl^ 3D cultures treated (TAM) or not (NO TAM) with tamoxifen and infected with lentiviruses carrying PTEN cDNA (FCIV-PTEN) or the empty vector (FCIV). Nuclei were evidenced with Hoechst staining. Data are from *n* = 3 experimental replicates (independent 3D cultures). Scale bars: 25 μm. (**B**) Representative images of SMAD2/3 staining on Cre:ER^+/−^;PTEN^fl/fl^ organoids treated (+LY) or not (-LY) with LY294002 10 µM and stimulated or not (UN) for 30 min with TGF-β with 10 ng/mL. Images of framed regions were magnified to evidence SMAD2/3 cellular localization. Data from *n* = 3 experimental replicates (independent organoid cultures). Scale bars: 25 μm. (**C**) Representative images of SMAD2/3 immunostaining on cells infected with a plasmid encoding a constitutively active AKT (CA-AKT) or the control plasmid (FCIV). Nuclei were evidenced with Hoechst staining. Scale bar 50 µm. (**D**) Representative images of SMAD2/3 immunostaining on Cre:ER^+/−^;PTEN^fl/fl^ organoids treated (TAM) or not (NO TAM) with tamoxifen and left or untreated (UN) or treated with 200 nM of Everolimus or Rapamycin. Nuclei were evidenced with Hoechst staining. Data from *n* = 3 experimental replicates (independent organoid cultures). Scale bar 50 µm. (**E**) Representative images of SMAD2/3 immunostaining on cells infected with a plasmid encoding a constitutively active AKT (CA-AKT) or the control plasmid (FCIV) and treated with 200 nM of Rapamycin or Everolimus or left untreated (UN). Nuclei were evidenced with Hoechst staining. Data from *n* = 3 experimental replicates (independent organoid cultures). Scale bar 50 µm.

**Figure 5 cancers-13-04990-f005:**
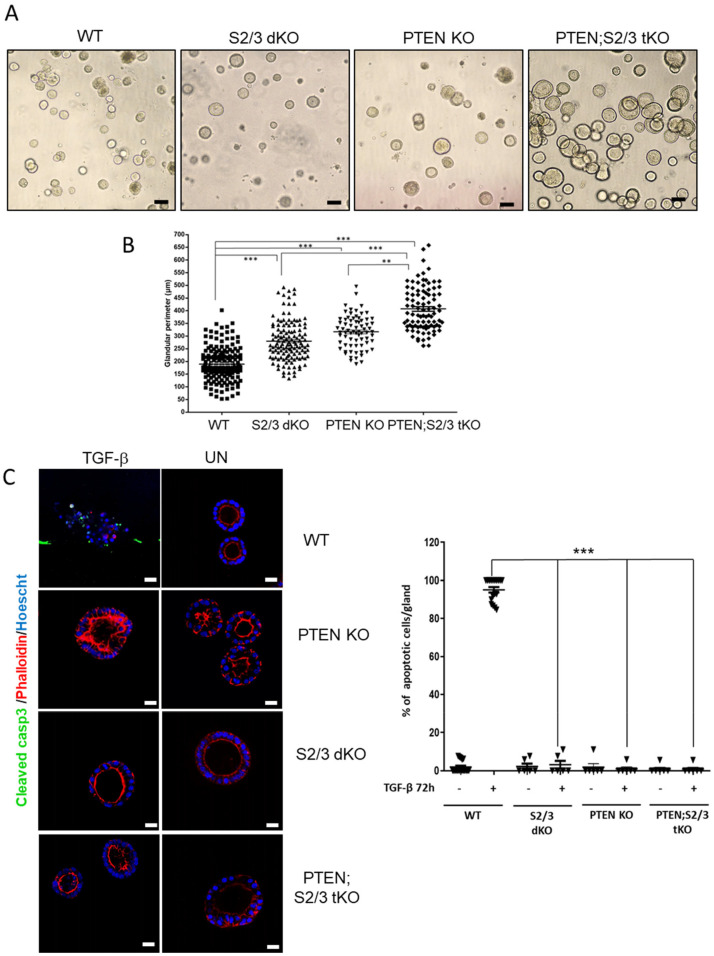
Nuclear SMAD2/3 constrains cell proliferation caused by PTEN deficiency. (**A**) Phase contrast pictures and (**B**) glandular perimeter measurements of Cre:ER^−/−^;PTEN^fl/fl^;SMAD2^fl/fl^;SMAD3^fl/fl^ without tamoxifen treatment (WT) or Cre:ER^+/−^;PTEN^fl/fl^;SMAD2^+/+^;SMAD3^+/+^ (PTEN knock-out, PTEN KO), Cre:ER^+/−^; Pten^+/+^;SMAD2^fl/fl^;SMAD3^fl/fl^ (SMAD2/3 double knock-out, S2/3 dKO) and Cre:ER^+/−^;PTEN^fl/fl^;SMAD2^fl/fl^;SMAD3^fl/fl^ (PTEN and SMAD2/3 triple knock-out, PTEN;S2/3 tKO) organoid cultures treated with tamoxifen (TAM). Values are mean and error bars represent mean ± s.e.m. ** *p* < 0.001, *** *p* < 0.0001 by *t*-test analysis. Data are shown for three independent experiments (each with *n* = 150 measured glands per condition). Scale bars: 100 μm. (**C**) Representative cleaved caspase-3 images (left) and quantification (right) of organoids with same genotypes as in (**A**) of treated with 10 ng/mL of TGF-β for 48 h or left untreated (UN). Cells were stained with phalloidin to evidence actin cytoskeleton and glandular morphology. To show apoptotic nuclear morphology nuclei were counterstained with Hoechst t. Scale bars: 25 μm. Values are mean and error bars represent mean ± s.e.m. *** *p* < 0.0001 by *t*-test analysis.

## Data Availability

Data Availability Statement: The datasets analyzed during the current study are available from the corresponding author upon reasonable request.

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
