# Peer review of "Endometrial PTEN Deficiency Leads to SMAD2/3 Nuclear Translocation"

_cancers, 2021, doi:10.3390/cancers13194990_

Round 1

Reviewer 1 Report

The authors report that the loss of PTEN leads to a constitutive SMAD2/3 nuclear translocation using an in vitro and in vivo model of endometrial carcinogenesis. Double PTEN/SMAD2/3 ablation results in a further increase of cell proliferation and enlarged endometrial organoids compared to those harboring single PTEN. The results suggests that nuclear translocation of SMAD2/3 triggered by PTEN deficiency has a tumor suppressive function and its loss leads to increased tumor growth in mouse endometrial organoids.

Overall, the manuscript is nice written and the data were logically presented. The data supports the conclusion solidly. Just minor revisions are needed to improve the data quality.

  1. The immunoblot of p-SMAD 2/3 on Fig. 1A has high background, please redo.
  2. The immunoblot of SMAD 2/3 on Fig. 1D is not reliable due to the black dots on the N fraction. Please redo.

Author Response

The authors report that the loss of PTEN leads to a constitutive SMAD2/3 nuclear translocation using an in vitro and in vivo model of endometrial carcinogenesis. Double PTEN/SMAD2/3 ablation results in a further increase of cell proliferation and enlarged endometrial organoids compared to those harboring single PTEN. The results suggests that nuclear translocation of SMAD2/3 triggered by PTEN deficiency has a tumor suppressive function and its loss leads to increased tumor growth in mouse endometrial organoids.

Overall, the manuscript is nice written and the data were logically presented. The data supports the conclusion solidly. Just minor revisions are needed to improve the data quality.

  1. The immunoblot of p-SMAD 2/3 on Fig. 1A has high background, please redo.
  2. The immunoblot of SMAD 2/3 on Fig. 1D is not reliable due to the black dots on the N fraction. Please redo.

We have substituted immunoblots by better quality ones.

Reviewer 2 Report

In this manuscript, Eritja et al. show that PTEN loss leads to nuclear translocation of Smad2/3 in endometrial organoids and that this is dependent on PI3K activity.

Original studies need to be referenced, not reviews. E.g. reference 14 and 15 should be replaced with the original studies.

Methods need to be provided in full.

Methods reference Figure O – what/where is this figure? Or is this meant to be FIGO?

Methods indicate human tissue samples were collected from grade 1, 2 and 3 EECs, but PTEN and SMAD2/3 IHC appear to have only been done on grade 3 cases (Figure 2B). What was staining like in grade 1 and 2 cases?

PTEN should be shown as a control in immunofluorescence experiments and Western blots.

The authors show that PTEN loss results in increased Smad2/3 expression – how is the expression of Smad2/3 regulated? Does PTEN bind to the Smad promoter or is this via post-transcriptional regulation?

Previous studies have looked at the mechanisms of SMAD2/3 translocation into the nucleus and these should be discussed.

The relevance of using SMAD2/3 as a biomarker of PTEN deficiency is questionable as PTEN status can be assessed relatively easily.

Figures 1A and 1D: Please add molecular weight markers to Western blots. Figure legend is missing.

Figure 1D: PTEN expression should be shown.

Figure 2A: Only data for TAM have been provided, data for NO TAM need to be provided (as stated in the figure legend) as well as H&E staining. Also need to add more information in the figure legend about what the black and white arrows indicate and magnification.

Figure 2B: Please add H&E staining. The graph on the right is very confusing. How were PTEN categories determined (i.e. what do 0, 1 and 2 mean)? Was SMAD2/3 staining quantified? Suggest plotting two graphs, one for PTEN vs nuclear SMAD and another for PTEN vs cytoplasmic SMAD.

Figure 3: What concentration of SB431542 was used?

Figure 3B: Insert size markers on the ladder. Data for TGFb treatment in TAM are missing. There appears to be a faint band in the IgG control of the last lane in the second panel (-501/-986, TAM, TGDb + SB431542).

Figure 3D: Include explanation of colours – is blue Hoechst and red protein of interest?

Figures 4A, D and F: Figure legends are missing.

Figure 4B: What concentration of LY294002 was used and is this sufficient to inhibit AKT phosphorylation in organoids? Data can be included as a Supplementary figure.

Also need to show that the concentrations of everolimus and rapamaycin used inhibit/reduce mTORC1 phosphorylation.

Figure 5C: Insert labels to explain what the top and bottom panels represent.

Author Response

In this manuscript, Eritja et al. show that PTEN loss leads to nuclear translocation of Smad2/3 in endometrial organoids and that this is dependent on PI3K activity.

  1. Original studies need to be referenced, not reviews. E.g. reference 14 and 15 should be replaced with the original studies.

As suggested by the reviewer, we have substituted the mentioned references for original studies. To avoid an over-sized list of references, in other cases we have references reviewers

  1. Methods need to be provided in full.

As requested by the reviewer, we provide an extended version of the methods.

  1. Methods reference Figure O – what/where is this figure? Or is this meant to be FIGO?

We apologize for the mistake, Figure O meant FIGO. We have made the correction.

  1. Methods indicate human tissue samples were collected from grade 1, 2 and 3 EECs, but PTEN and SMAD2/3 IHC appear to have only been done on grade 3 cases (Figure 2B). What was staining like in grade 1 and 2 cases?

  1. PTEN should be shown as a control in immunofluorescence experiments and Western blots.

In all western blots we provide PTEN immunoblots as a control. In figure 1D, the control for PTEN expression is the same as in Figure 1A. For immunoblots shown in Figure 1A and 1D (1E in the revised version of the manuscript), organoids were collected and split fractions, one was used for total protein extraction and the other one for nuclear and cytosolic extraction. The reason for performing PTEN immunoblot only in Figure 1A is the availability of protein. The amount of total protein obtained from organoid cultures is very low, and the amount of protein extracted from nuclear and cytosolic lysates is extremely even lower. This short amount of protein limits the number of immunoblots that can be performed. For this reason in nuclear and cytosolic extracts, we only performed the more important controls, like histone and LDH to control the purity of nuclear and cytosolic fractions, respectively. We clarify this point in the results section (page 5):

For this purpose,  PTEN Cre:ER+/-;Ptenfl/fl organoids treated or not with tamoxifen to induce PTEN excision were collected and split in two fractions. The first fraction was used for total protein extraction and subjected to analysis by western blot. The second fraction was used for cytosolic and nuclear protein extraction”

In case of immunofluorescence, none of the antibodies we have tested gives enough sensitivity and/or specificity for PTEN immunofluorescence. However, we would like to mention that PTEN loss induced by tamoxifen addition is routinely checked by western blot using parallel cultures incubated exactly under the same conditions.

  1. The authors show that PTEN loss results in increased Smad2/3 expression – how is the expression of Smad2/3 regulated? Does PTEN bind to the Smad promoter or is this via post-transcriptional regulation?

To address this point we have performed an RT-PCR analysis of SMAD2, SMAD3 and SMAD4. As we show in the new Figure 1B of the revised manuscript, loss of PTEN does not cause any significant modification of SMADs expression, suggesting that the increase of SMAD protein might be due to its post-transcriptional regulation. To this regard, it has been reported that Akt can regulate SMADs stability (reviewed in Pinglong Xu et al., FEBS letters, 2012). We have also added a discussion of this point in Discussion section.

  1. Previous studies have looked at the mechanisms of SMAD2/3 translocation into the nucleus and these should be discussed. 

In the discussion section of the manuscript, we cite and discuss reports that have previously addressed the role of PI3K/Akt signaling in SMAD2/3 translocation to the nucleus:

“Mechanistically, the regulation of SMADs activation and their nuclear translocation by the PI3K/AKT signaling pathway is still controversial and opposing effects of PI3K/AKT activation on SMAD activity and localization have been observed. On the one hand, it has been reported that AKT can directly interact with SMAD3 inhibiting its nuclear translocation and activation[22,23]. Also, activation of PI3K/AKT signalling by IGF-1 suppresses SMAD3 activation in prostate cells[44] On the other hand, it has been also demonstrated that enhanced PI3K/AKT signaling triggers SMAD activation in several cell types with different cellular outcomes. In keratinocytes, loss of PTEN increases TGFβ–mediated Invasion with enhanced SMAD3 transcriptional activity[45]. In the kidney, PTEN loss initiates tubular dysfunction via SMAD3 dependent fibrotic responses[46]. Prostates from PTEN deficient mice display increased phosphorylation and activation of SMAD3 and SMAD4[25].”

However, if the reviewer feels that this point of the discussion should be further extended, we will have no objection to do so.

  1. The relevance of using SMAD2/3 as a biomarker of PTEN deficiency is questionable as PTEN status can be assessed relatively easily.

We completely agree with the reviewer that PTEN can routinely and easily assessed by immunohistochemistry. However, we would like to emphasize that the correlation between the lack of PTEN and nuclear SMAD3 is restricted to Grade III EC, which are poorly differentiated and considered as high-risk EC that often spread to other parts of the body. We clarified the potential use of SMAD2/3 as biomarker downstream of PTEN deletion in the discussion section.

  1. Figures 1A and 1D: Please add molecular weight markers to Western blots. Figure legend is missing.

Molecular weight markers have been added to Figure 1A and 1D (1E in revised version) immunblots. Weight markers are also included in the file containing uncropped western blots.

We have added Figure legends corresponding to Figures 1A and 1D (corresponding to Figures 1A and 1E on the revised version of the manuscript)

  1. Figure 1D: PTEN expression should be shown.

As we mentioned above, PTEN expression corresponding to Figure 1D (Figure 1E in the revised version) is shown in Figure 1A. We clarified this point in the revised version of the manuscript.

  1. Figure 2A: Only data for TAM have been provided, data for NO TAM need to be provided (as stated in the figure legend) as well as H&E staining. Also need to add more information in the figure legend about what the black and white arrows indicate and magnification.

We apologize for this mistake in the legend. Due to image size and space reasons in Figure 2A we only show images from tamoxifen treated mice (TAM), but not images from tamoxifen untreated mice (NO TAM). For this reason NO TAM have been deleted from figure legend. We provide control images showing the increase of nuclear SMAD2/3 in PTEN-deficient glands using tamoxifen-treated (TAM) and non-treated littermates (NO TAM) in supplementary Figure 1B.

We have included information about the meaning of black and white arrows and we have included the magnification of images.

Figure 2B: Please add H&E staining. The graph on the right is very confusing. How were PTEN categories determined (i.e. what do 0, 1 and 2 mean)? Was SMAD2/3 staining quantified? Suggest plotting two graphs, one for PTEN vs nuclear SMAD and another for PTEN vs cytoplasmic SMAD

 In the Methods section of the revised version of the manuscript, we have tried to clarify this confusing point about TMA quantification with the following information:

“In case of TMA evaluation, immunohistochemical evaluation was done after examining the two different tumor cylinders from each case. PTEN immunoreactivity was scored as follows: 2 for highly expressing cylinders, 1 for moderately expressing cylinders and 0 for cylinders completely lacking PTEN expression. For evaluation of SMAD2/3 for cytosolic and nuclear staining intensity, cylinders were scored as follows: n, for cylinders showing only nuclear expression; c, for cylinders showing only cytoplasmic expression; nc, for cylinders showing both nuclear and cytosolic expression. The reliability of such score for interpretation of immunohistochemical staining in EC TMAs has been shown previously [55, 56].

The cylinders lacking PTEN expression (0) are considered the ones harboring PTEN mutations that lead to its loss of expression (and therefore comparable to the loss of expression caused by tamoxifen-induced deletion in PTEN knock-out mouse model).

We have also substituted the graph shown in Figure 2B by a new one showing nuclear and cytoplasmic quantification of SMAD2/3 separated by histological grades I, II, III. We have also clarified in the text that the correlation between PTEN loss and SMAD2/3 nuclear translocation is observed in Grade III EC.

  1. Figure 3: What concentration of SB431542 was used?

The dose of SB431542 is 10 µM. We have included this information in the corresponding Figure legend.

  1. Figure 3B: Insert size markers on the ladder. Data for TGFb treatment in TAM are missing. There appears to be a faint band in the IgG control of the last lane in the second panel (-501/-986, TAM, TGDb + SB431542).

We have inserted the label for size markers surrounding PCR band.

The objective of the experiments shown in Figure 3 was to demonstrate that the constitutive translocation of SMAD3 downstream of PTEN deletion was independent of TGFβ receptor. For this reason, the experiment was performed only with PTEN KO cells (TAM) in absence of TGFβ. Wild type cells stimulated with TGFβ were used as a control of SB431542 effects. As it happens with western blots and protein, it is difficult to obtain enough amount of DNA to perform ChIP experiments from organoid cultures obtained from KO mice, making impossible to have all conditions. For this reason, we decided to include the control of SB431542 effects using wild type cells stimulated with TGFβ and dispense with TGFβ stimulated PTEN KO.

Figure 3D: Include explanation of colours – is blue Hoechst and red protein of interest?

An explanation of colours have been included in the Figure legend. As indicated by the reviewer blue correspond to Hoescht staining and red to the protein of interest (TGFβRI and II).

  1. Figures 4A, D and E: Figure legends are missing.

Again, we apologize for such neglet. Figure legends corresponding to Figures 4A, d and E have been included in the revised version of the manuscript.

  1. Figure 4B: What concentration of LY294002 was used and is this sufficient to inhibit AKT phosphorylation in organoids? Data can be included as a Supplementary figure.

The doses of LY294002 was 10 µM. We have included the concentration in Figure 4 legend of the revised version of the manuscript. As we have been using LY294002 for long time in  previous studies from our laboratory, we already know that this dose is effective to inhibit Akt phosphorylation. However, as suggested by the reviewer, we provide a western blot showing the LY294002 blocks Akt phosphorylation. This data have been included as Supplementary Figure 2B.

  1. Figure 5C: Insert labels to explain what the top and bottom panels represent.

We have included the labels UN and TGFβ for the images shown in Figure 5C.

Reviewer 3 Report

This manuscript presents the role of PTEN and SMAD interaction in tumourigenesis of endometrial cancer. Generally, the methodological approach sounds great. Developing of in vitro 3D models and the in vivo models are the strong side of this investigation. However, there are several limitations in this study.

  • The authors did not mention the technical and biological triplicates of the study. Most of the data are qualitative and it is not known whether the experiments are significant or not.
  • The authors did not mention any area that address the optimization of the antibodies. It is crucial antibody optimization as there could be likely cross reactivity of the antibodies can lead to wrong conclusion.
  • In this study the authors analysed patient samples for PTEN expression and SMAD2/2 expression and mentioned only GIII tumours showed deficiency of PTEN. Did this show these patients have mutation of PTEN. It would be interesting to demonstrate the status of PTEN mutation and expression to give the conclusion PTEN loss equates with nuclear expression SMAD2/3. To be conclusive the authors have to correlate the PTEN expression/SMAD2/3 expression ratio with clinicopathologic and clinical outcome in this cohort of patients
  • It would be interesting if the authors demonstrated a data that show differential gene expression upon KO of PTEN and SMAD2/3
  • The conclusion given in this investigation seems strong and still there is gap of experiments. Although it is mentioned that a functional study of proliferation in previous study with KO of SMAD2/3 it is not adequate. Authors need to present more functional data on proliferation, migration and colony formation etc.
  • The SMAD signalling is know for EMT and the authors need to clarify under what condition is SMAD playing a role as tumour suppressor.
  • The authors need to support their data by performing the publicly available data for example TCGA if there is any association PTEN mutation with SMAD2/3 expression.

Author Response

This manuscript presents the role of PTEN and SMAD interaction in tumourigenesis of endometrial cancer. Generally, the methodological approach sounds great. Developing of in vitro 3D models and the in vivo models are the strong side of this investigation. However, there are several limitations in this study.

  1. The authors did not mention the technical and biological triplicates of the study. Most of the data are qualitative and it is not known whether the experiments are significant or not.

In the Figure legends of the revised manuscript, we have included the number of replicates made in each experiment.

We have included quantifications of most immunofluorescence and immunohistochemistry studies. Data corresponding to the nuclear translocation experiments is shown in the new Supplementary Figure 3 of the revised version of the manuscript and cited in the text.

Quantification of nuclear vs citoplasmatic staining of SMAD2/3 in PTEN mice is shown in Supplementary Figure 1.

We also provide quantification caspase-3 immunofluorescence shown in Figure 5C.

  1. The authors did not mention any area that address the optimization of the antibodies. It is crucial antibody optimization as there could be likely cross reactivity of the antibodies can lead to wrong conclusion.

In the revised version of the manuscript, we have extended methodology section including the protocols used for immunofluorescence and immunohistochemistry. The specificity of the ost important antibodies used in this study have been tested using the knock-out mice. For instance, in Figure 1A we show that there is a complete lack of PTEN signal in Cre:ER+/-;Pten fl/fl  treated with tamoxifen (PTEN KOs). Similarly, there e is a complete loss of PTEN immunostaining in PTEN-deficient endometrial glands (Figure 2A and Supplementary Figure 1B).Our main concern in the present study was the specificity of SMAD2/3 antibody, as most of our results are supported by SMAD2/3 immunofluorescence. For this reason, we tested SMAD2/3 antibody specificity using organoid cultures obtained from SMAD2/3 double knock-out mice. As we shown in Supplementary Figure 2, there is a complete lack of SMAD2/3 staining in double knock-out mice. We think that the antibody test using the knock-out mice is the best proof for antibody specificity.  

  1. In this study the authors analyzed patient samples for PTEN expression and SMAD2/3 expression and mentioned only GIII tumours showed deficiency of PTEN. Did this show these patients have mutation of PTEN. It would be interesting to demonstrate the status of PTEN mutation and expression to give the conclusion PTEN loss equates with nuclear expression SMAD2/3. To be conclusive the authors have to correlate the PTEN expression/SMAD2/3 expression ratio with clinicopathologic and clinical outcome in this cohort of patients.

In the Methods section of the revised version of the manuscript we have tried to clarify this confusing point about TMA quantification with the following information:

“In case of TMA evaluation, immunohistochemical evaluation was done after examining the two different tumor cylinders from each case. PTEN immunoreactivity was scored as follows: 2 for highly expressing cylinders, 1 for moderately expressing cylinders and 0 for cylinders completely lacking PTEN expression. For evaluation of SMAD2/3 for cytosolic and nuclear staining intensity, cylinders were scored as follows: n, for cylinders showing only nuclear expression; c, for cylinders showing only cytoplasmic expression; nc, for cylinders showing both nuclear and cytosolic expression. The reliability of such score for interpretation of immunohistochemical staining in EC TMAs has been shown previously[55, 56]”

The cylinders lacking PTEN expression (0) are considered the ones harboring PTEN mutations that lead to its loss of expression (and therefore comparable to the loss of expression caused by tamoxifen-induced deletion in PTEN knock-out mouse model).

We have also substituted the graph shown in Figure 2B by a new one showing nuclear and cytoplasmic quantification of SMAD2/3 separated by histological grades I, II, III. We have also clarified in the text that the correlation between PTEN loss and SMAD2/3 nuclear translocation is observed in Grade III EC.

  1. It would be interesting if the authors demonstrated a data that show differential gene expression upon KO of PTEN and SMAD2/3

As also requested by reviewer 2, in the Figure 1B of the revised version of the manuscript, we show an RT-PCR analysis that demonstrates that loss PTEN does not change relative mRNA levels of SMAD2, SMAD3 or SMAD4.

  1. The conclusion given in this investigation seems strong and still there is gap of experiments. Although it is mentioned that a functional study of proliferation in previous study with KO of SMAD2/3 it is not adequate. Authors need to present more functional data on proliferation, migration and colony formation etc.

We have not observed any difference in cell migration, glandular morphology or EMT between PTEN knock-out and triple PTEN;SMAD2/3 organoids. We have only found the increase in cell proliferation and the resistance to apoptosis that we show in Figure 5. Regarding this concern, we would like to point out that the main objective of the present work is to demonstrate that SMAD2/3 is a downstream target of PTEN deficiency. However, we have considered interesting to add few functional studies in the last figure.

We are currently conducting a long term study in which we are phenotyping PTEN;SMAD2/3 triple knock-out. In this study, we will address the function of PTEN and SMAD2/3 in endometrial, prostate and thyroid carcinogenesis. We think, in the future, that this study will provide further information about the consequences of PTEN and SMAD2/3 deletion in tumoral phenotype.  

  1. The SMAD signalling is know for EMT and the authors need to clarify under what condition is SMAD playing a role as tumour suppressor.

We have tried to clarify this point. In summary, our results support a tumor-suppressive function of SMAD signalling in the endometrium, but a tumor promoting in PTEN deficient endometrium. The tumor promoting function is supported by an increased glandular perimeter and a suppression of apoptosis triggered by TGF-β. We completely agree with the reviewer that SMAD signalling triggers EMT in some tumoral types. However, in endometrial organoids, we have not found any sign of EMT in PTEN deficient or double PTEN/SMAD2/3 deficient mice.

This clarification has been included in the last paragraph of the discussion.

  1. The authors need to support their data by performing the publicly available data for example TCGA if there is any association PTEN mutation with SMAD2/3 expression.

As we shown in new Figure 1B of the revised manuscript (also requested by reviewer 2), loss of PTEN does not affect SMADs mRNA levels, suggesting that PTEN does not transcriptionally regulate SMADs expression. Therefore, we do not expect to find differences in the expression of SMADs in PTEN mutated human samples.

Round 2

Reviewer 2 Report

I thank the authors for their explanations and modifications they have made. I just have some text edits that need to be made.

ChIP methods section 2.6 line 4 states “lysate was sonicated the times” – please state how many times the lysate was sonicated.

As SMAD RNA levels are similar in the presence or absence of PTEN (Figure 1B), this suggests that PTEN deletion does not affect SMAD transcription and that instead, post-transcriptional mechanisms are involved. In results section 3.1 line 14, the authors state that the “increase of protein is not caused by post-transcriptional mechanisms”, whereas the results suggest the opposite, i.e. that increase of SMAD2/3 protein appears to be due to post-transcriptional mechanisms. Please correct this sentence.

Please ensure that the last sentence in the Conclusion is consistent with what is stated in the Discussion (i.e. SMAD2/3 as a biomarker of PTEN deficiency only in grade 3 EC).

Figure 3B: Please insert size markers for the DNA ladder.

Sup Fig 2B shows a reduction in, not inhibition of, pAKT with LY294002. Please correct the text in results section 3.3 line 11.

Please check labels in Supplementary Figure 3 as some Figures have changed (e.g. Figure 1B is now 1C).

Author Response

I thank the authors for their explanations and modifications they have made. I just have some text edits that need to be made.

ChIP methods section 2.6 line 4 states “lysate was sonicated the times” – please state how many times the lysate was sonicated.

 We have corrected the sentence, “the” has been substituted by “ten”

As SMAD RNA levels are similar in the presence or absence of PTEN (Figure 1B), this suggests that PTEN deletion does not affect SMAD transcription and that instead, post-transcriptional mechanisms are involved. In results section 3.1 line 14, the authors state that the “increase of protein is not caused by post-transcriptional mechanisms”, whereas the results suggest the opposite, i.e. that increase of SMAD2/3 protein appears to be due to post-transcriptional mechanisms. Please correct this sentence.

 We apologize for this confusion in the text. We really want to say that “increase of protein is caused by post-transcriptional mechanisms”. We have made the correction

Please ensure that the last sentence in the Conclusion is consistent with what is stated in the Discussion (i.e. SMAD2/3 as a biomarker of PTEN deficiency only in grade 3 EC).

 We have clarified that SMAD2/3 may be a biomarker in Grade III EC.

Figure 3B: Please insert size markers for the DNA ladder.

 The size markers for DNA ladder bands surrounding PCR products have been included

Sup Fig 2B shows a reduction in, not inhibition of, pAKT with LY294002. Please correct the text in results section 3.3 line 11.

We have corrected the sentence. 

Please check labels in Supplementary Figure 3 as some Figures have changed (e.g. Figure 1B is now 1C).

We have corrected the labels.

Reviewer 3 Report

First, I would like to thank the authors for their comprehensive revising the manuscript and submitting for review. The manuscript looks much better than the previous version. When I go through the paper again and I found some contradicting concepts and then I started to review critically.

The Title of the manuscript “Endometrial PTEN Deficiency Leads to Tumor Suppressive 

SMAD2/3 Nuclear Translocation” may need rewording as the current title creates confusion. I suggest removing the “tumour suppressive (adjective phrase) from the title and just Endometrial PTEN Deficiency promotes to SMAD2/3 Nuclear Translocation.  

Page 1 Abstract:  

Abstract: TGF-β has a dichotomous function, acting as tumor suppressor in premalignant cells but as a tumor promoter for cancerous cells. These contradictory functions of TGF-β are caused by different cellular contexts, including both intracellular and environmental determinants. The TGF-β/SMAD and the PI3K/PTEN/AKT signal transduction pathways have an important role in the regulation of epithelial cell homeostasis and perturbations in either of these two pathways contribute to endometrial carcinogenesis. “”We have previously demonstrated that genetic ablation of either PTEN or SMAD2/3 displays tumor suppressive functions in the endometrium” (not true) it sound like knockout of PTEN enhance tumour suppression but the fact is the opposite. ……genetic ablation of either gene results in sustained activation of PI3K/AKT signaling that suppresses TGF-β-induced apoptosis and enhances cell proliferation of mouse endometrial cells. This concept is contradicting to each other. It is proven that PTEN is tumour suppressor, and its deficiency promotes tumorigeneses in endometrial cancer 

However, the molecular and cellular effects of PTEN deficiency on TGF-β/SMAD2/3 signaling remain controversial. Here, using an in vitro and in vivo model of endometrial carcinogenesis, we have demonstrated that loss of PTEN leads to a constitutive SMAD2/3 nuclear translocation. To ascertain the function of nuclear SMAD2/3 downstream of PTEN deficiency, we analyzed the effects of double deletion PTEN and SMAD2/3 in mouse endometrial organoids. Double PTEN/SMAD2/3 ablation results in a further increase of cell proliferation and enlarged endometrial organoids compared to those harboring single PTEN, suggesting that nuclear translocation of SMAD2/3 triggered by PTEN deficiency has a tumor suppressive function (not true) and its loss leads to increased tumor growth. This concept is also contradicting to each other

Page 12 Introduction: TGF-β type II receptor (TβRII) that, in turn, interacts with the TGF-β type I receptor (TβRI). TβRII and TβRI are not standard the abbreviation, please correct as TGFβRII and TGFβRI respectively throughout the manuscript. 

Page 2 Introduction Paragraph3: PTEN (phosphatase and tensin homolog deleted on chromosome 10) is a phosphatase that opposes PI3K activity by dephosphorylating phosphatidylinositol-3,4,5-trisphosphate (PIP3) to phosphatidylinositol-4,5-trisphosphate  (PIP2)[13]. Do you mean located? You need to revise the word. 

Method page4 “3% Matrigel” from our experience to organoid culture and published paper 3% seems too low for organoid culture. Is this a typo error to indicate 30% or your lab protocol is 3% Matrigel.

Results page 8 Figure 1 image C and D, some of the IF confocal images have scale bar but some of them do not. Please add scale bar to all confocal IF images. Similarly, please add scale bar for the IHC images in Figure 2 and Figure 3 to both IF and IHC panels. In Figure 3 C, please align the negative IHC control image either with the top panel or the bottom panel.

In Figure 4 the scale bar is missed from some of the panels, please be consistent to present the data and add scale bar to those IF image panels that lack scale bar. The image alignment is also looking a bit weird as the magnified view of the panels are not inline with their respective low magnification view. Could you please crop a larger image and align with respective inset for readability and data visualization? Again please add scale bar to all panels in Figure 5.

Finally, the concluding sentence “Our data strongly suggest that SMAD2/3 has tumor suppressive functions when it is translocated to nucleus by either TGF-b stimulation or by PTEN deficiency” seems contradicting to the finding. All the data described tells us deletion of PTEN promotes localization of SMAD2/3 and this increased organoid number (in vitro and glandular perimeter in in vivo). If that is the case how PTEN deficiency triggers tumour suppression and this is against the well know function of PTEN.

Minor comments

There are occasional inappropriate words usage throughout the manuscript that needs revision and correction.

Author Response

First, I would like to thank the authors for their comprehensive revising the manuscript and submitting for review. The manuscript looks much better than the previous version. When I go through the paper again and I found some contradicting concepts and then I started to review critically.

The Title of the manuscript “Endometrial PTEN Deficiency Leads to Tumor Suppressive 

SMAD2/3 Nuclear Translocation” may need rewording as the current title creates confusion. I suggest removing the “tumour suppressive (adjective phrase) from the title and just Endometrial PTEN Deficiency promotes to SMAD2/3 Nuclear Translocation.  

As suggested by the reviewer, we have changed the tittle.

Page 1 Abstract:  

Abstract: TGF-β has a dichotomous function, acting as tumor suppressor in premalignant cells but as a tumor promoter for cancerous cells. These contradictory functions of TGF-β are caused by different cellular contexts, including both intracellular and environmental determinants. The TGF-β/SMAD and the PI3K/PTEN/AKT signal transduction pathways have an important role in the regulation of epithelial cell homeostasis and perturbations in either of these two pathways contribute to endometrial carcinogenesis. “”We have previously demonstrated that genetic ablation of either PTEN or SMAD2/3 displays tumor suppressive functions in the endometrium” (not true) it sound like knockout of PTEN enhance tumour suppression but the fact is the opposite. ……genetic ablation of either gene results in sustained activation of PI3K/AKT signaling that suppresses TGF-β-induced apoptosis and enhances cell proliferation of mouse endometrial cells. This concept is contradicting to each other. It is proven that PTEN is tumour suppressor, and its deficiency promotes tumorigeneses in endometrial cancer 

We completely agree with the reviewer that PTEN is a tumor suppressor. We apologize for this apparent contradiction. We have modified the abstract text to try to clarify that PTEN is really a tumor suppressor gene and its loss promotes tumorigenesis. 

However, the molecular and cellular effects of PTEN deficiency on TGF-β/SMAD2/3 signaling remain controversial. Here, using an in vitro and in vivo model of endometrial carcinogenesis, we have demonstrated that loss of PTEN leads to a constitutive SMAD2/3 nuclear translocation. To ascertain the function of nuclear SMAD2/3 downstream of PTEN deficiency, we analyzed the effects of double deletion PTEN and SMAD2/3 in mouse endometrial organoids. Double PTEN/SMAD2/3 ablation results in a further increase of cell proliferation and enlarged endometrial organoids compared to those harboring single PTEN, suggesting that nuclear translocation of SMAD2/3 triggered by PTEN deficiency has a tumor suppressive function (not true) and its loss leads to increased tumor growth. This concept is also contradicting to each other

Again, we apologize for the contradiction. We have made a text correction to clarify the role of PTEN in EC. 

Page 12 Introduction: TGF-β type II receptor (TβRII) that, in turn, interacts with the TGF-β type I receptor (TβRI). TβRII and TβRI are not standard the abbreviation, please correct as TGFβRII and TGFβRI respectively throughout the manuscript. 

We have made the correction as indicated by the reviewer.

Page 2 Introduction Paragraph3: PTEN (phosphatase and tensin homolog deleted on chromosome 10) is a phosphatase that opposes PI3K activity by dephosphorylating phosphatidylinositol-3,4,5-trisphosphate (PIP3) to phosphatidylinositol-4,5-trisphosphate  (PIP2)[13]. Do you mean located? You need to revise the word. 

We do not really know if we understand the question properly. The abbreviation PTEN stands for the complete name of the protein that, as far as we know is: “Phosphatase and tensin homolog deleted (no located) on chromosome 10”.  

Method page4 “3% Matrigel” from our experience to organoid culture and published paper 3% seems too low for organoid culture. Is this a typo error to indicate 30% or your lab protocol is 3% Matrigel.

We use a concentration of 3% of Matrigel dissolved in culture medium, but the cells are layered on 100% Matrigel coated tissue culture wells. We have clarified this point in the Methods section.

Results page 8 Figure 1 image C and D, some of the IF confocal images have scale bar but some of them do not. Please add scale bar to all confocal IF images. Similarly, please add scale bar for the IHC images in Figure 2 and Figure 3 to both IF and IHC panels. In Figure 3 C, please align the negative IHC control image either with the top panel or the bottom panel.

We have included scale bars and aligned the figure as indicated by the reviewer.

In Figure 4 the scale bar is missed from some of the panels, please be consistent to present the data and add scale bar to those IF image panels that lack scale bar. The image alignment is also looking a bit weird as the magnified view of the panels are not inline with their respective low magnification view. Could you please crop a larger image and align with respective inset for readability and data visualization? Again please add scale bar to all panels in Figure 5.

We have included scale bars and aligned the magnifications as indicated by the reviewer.

Finally, the concluding sentence “Our data strongly suggest that SMAD2/3 has tumor suppressive functions when it is translocated to nucleus by either TGF-b stimulation or by PTEN deficiency” seems contradicting to the finding. All the data described tells us deletion of PTEN promotes localization of SMAD2/3 and this increased organoid number (in vitro and glandular perimeter in in vivo). If that is the case how PTEN deficiency triggers tumour suppression and this is against the well know function of PTEN.

We completely agree with the reviewer that PTEN is a tumor suppressor and its deficiency triggers EC tumorigenesis. We apologize for the confusion in the text. As we did in the abstract, we have modified the text of the conclusion to make clear that PTEN is a tumor suppressor. Indeed, PTEN loss leads to an increase of organoid size and triggers SMAD2/3 nuclear translocation. When PTEN and SMAD2/3 and organoid size is further increased, indicating that SMAD2/3 nuclear translocation constrains tumorigenesis triggered by PTEN loss.

Minor comments

There are occasional inappropriate words usage throughout the manuscript that needs revision and correction.

We have reviewed word usage to try to improve it.